# Challenges for Optimization of Reverse Shoulder Arthroplasty Part I: External Rotation, Extension and Internal Rotation

**DOI:** 10.3390/jcm12051814

**Published:** 2023-02-24

**Authors:** Stefan Bauer, William G. Blakeney, Allan W. Wang, Lukas Ernstbrunner, Jean-David Werthel, Jocelyn Corbaz

**Affiliations:** 1Service d’Orthopédie et de Traumatologie, Chirurgie de l’Épaule, Ensemble Hospitalier de la Côte, 1110 Morges, Switzerland; 2Medical School, University of Western Australia, 35 Sterling Highway, Perth, WA 6009, Australia; 3Department of Orthopaedic Surgery, Royal Perth Hospital, Perth, WA 6000, Australia; 4Department of Orthopaedic Surgery, Royal Melbourne Hospital, 300 Grattan Street, Parkville, VIC 3050, Australia; 5Department of Biomedical Engineering, University of Melbourne, Parkville, VIC 3010, Australia; 6Service d’Orthopédie et de Traumatologie, Hôpital Ambroise Paré, 9 Avenue Charles de Gaulle, 92100 Boulogne-Billancourt, France; 7Service d’Orthopédie et de Traumatologie, Centre Hospitalier Universitaire Vaudois, 1011 Lausanne, Switzerland

**Keywords:** reverse shoulder arthroplasty, optimization, external rotation, extension, internal rotation

## Abstract

A detailed overview of the basic science and clinical literature reporting on the challenges for the optimization of reverse shoulder arthroplasty (RSA) is presented in two review articles. Part I looks at (I) external rotation and extension, (II) internal rotation and the analysis and discussion of the interplay of different factors influencing these challenges. In part II, we focus on (III) the conservation of sufficient subacromial and coracohumeral space, (IV) scapular posture and (V) moment arms and muscle tensioning. There is a need to define the criteria and algorithms for planning and execution of optimized, balanced RSA to improve the range of motion, function and longevity whilst minimizing complications. For an optimized RSA with the highest function, it is important not to overlook any of these challenges. This summary may be used as an aide memoire for RSA planning.

## 1. Introduction

Reverse shoulder arthroplasty (RSA) has demonstrated good clinical outcomes, especially for active forward elevation. Studies comparing anatomic total shoulder arthroplasty (aTSA) and RSA in patients with an intact rotator cuff, however, have shown a greater postoperative range of motion (ROM) with aTSA [1]. In particular, sufficient internal and external rotation are more difficult to achieve after RSA.

Efforts to optimize ROM in RSA have typically been focused on impingement-free adduction (ADD) and extension (EXT) to prevent notching [2,3]. Scapular notching is a concern after RSA [4,5] and the most frequent complication [6]. External rotation (ER) with the arm at side and extension (EXT) seems to be a key player in the notching phenomenon [7]. On the other hand, physiological ER is most useful with the arm in an elevated position to reach the head and to position the hand in space. For this combined movement, the RSA configuration should respect and maintain sufficient subacromial and coracohumeral space [8,9], although dynamic scapulothoracic motion is able to compensate and contribute significantly to enable motion with the arm in an elevated position. The loss of active ER with the clinical manifestation of a hornblower sign is associated with atrophy and fatty infiltration of the teres minor and has been identified as a predictor of poor results after RSA [5]. Improving the internal rotation (IR) as a combined movement (EXT + IR) [10] to place the arm back has long been overlooked but has recently been given more attention [11]. The restoration of IR with RSA remains a complex multifactorial challenge of passive and active factors.

Many studies have looked at certain aspects of these challenges in isolation. These studies include computer simulations and biomechanical cadaveric studies, as well as clinical studies. This comprehensive review looks at all the current evidence for the optimization of reverse shoulder arthroplasty function and synthesizes it into the following sections. Part I looks at (I) external rotation and extension and (II) internal rotation. In part II, we focus on (III) the conservation of sufficient subacromial and coracohumeral space, (IV) scapular posture and (V) moment arms and muscle tensioning.

## 2. External Rotation and Extension

In RSA, regaining ER is a technical challenge. As the humerus goes into ER, it can impinge on the scapula. Scapular notching was first described in a case report by Nyffeler at al. [4] and classified shortly after by Sirveaux with the publication of mid-term results of a series of Grammont RSA performed in the nineties [5]. It is the most common complication after RSA with the Grammont design [6] and has been attributed to a medialized glenosphere (GS) and a 155° neck-shaft angle (NSA) [12]. At first, it was believed to be caused by impingement in ADD [13] before observations of notching in ER were also found [14]. With the advent of 3D CT modeling, a combined biomechanical cadaver study revealed repeated flexion (FLEX), EXT and ER with the arm at the side as the main cause of notching, a mechanism termed friction-type impingement [7]. Notching has been shown to be associated with adverse clinical outcomes, and it was therefore recommended to avoid a high base plate position and superior tilt. Simovitch et al. found significantly poorer clinical outcomes associated with notching, which was more prevalent in patients of short stature, lower weight and body mass index [15], and Spiry et al. showed the association between grade IV notching and aseptic glenoid loosening [16].

Middernacht et al. were the first to analyze the scapula and glenoid anatomy and their relevance for RSA biomechanics [17]. They analyzed cadaveric dry bone scapulae and found that the lateral border of the scapula and infraglenoid tubercle were located posterior to the longitudinal axis of the glenoid. They described this relationship as “posterior offset of the lateral border of the scapula in the glenoid plane”, as shown in Figure 1.

In keeping with the findings of Middernacht et al., it was shown in a gender-matched computer model that the posteroinferior distance from the scapular margin to the GS, termed posteroinferior relevant scapular neck offset (pRSNO), was always and significantly smaller (<0.001) than the anteroinferior distance, termed anteroinferior relevant scapular neck offset (aRSNO) [18]. Figure 2 shows the location of the pRSNO and aRSNO in RSA (Figure 2). A shorter pRSNO showed a strong correlation with decreased ERO and EXT (Figure 3A,B), with a potential for friction-type impingement. Lateralization (LAT) combined with an eccentric GS increased the pRSNO and was the best configuration for rigid body motion in this computer model (Figure 3C). For this study, a semi-inlay 145° implant was chosen as a constant, which had previously been shown to provide the best compromise for motion with the arm at the side and in an elevated position [19]. Arenas et al. found that glenoid LAT was the most effective method to increase global ROM. A semi-inlay 145° design with 4 mm LAT and a GS with 2 mm eccentricity was defined as the middle ground for RSA for Walch A1 glenoids [20] used in their computer study; however, the large potential for dynamic scapulothoracic contribution to ROM with the arm in an elevated position was not taken into account in their isolated glenohumeral computer model.

### 2.1. Importance of Notching and How to Prevent It?

Radiographic notching has been shown to be associated with poorer clinical outcomes [15] and is probably associated with mechanical impingement in IR and ER, causing limitations in passive axial rotation [21] and long-term glenoid loosening [16]. Werthel et al. provided an analysis of the lateralization of RSA implants on the market and broke it down into humeral- and glenoid-sided LAT [22]. The benefits and risks of glenoid, humeral-sided and combined LAT have been discussed in recent years [22,23,24]. A number of authors have demonstrated that humeral LAT cannot reduce notching. However, although decreasing the NSA to 145° or 135° has very minimal influence on humeral LAT, it does reduce the risk of notching by tilting the polyethylene insert away from the scapular pillar. According to the authors, the notching phenomenon may be influenced by four factors:(1)Location of center of rotation (COR) of GS relative to the glenoid bone;(2)Humeral NSA;(3)Shape of the scapular pillar;(4)Shape of the scapular neck (which can be elongated by glenoid LAT).

As a fifth factor, we would add:(5)Distance of the scapular pillar in relation to the posteroinferior extent of GS [18].

Factor (5) is the definition of pRSNO, which can be increased in the most efficient way by combining glenoid LAT, GS eccentricity and increased GS overhang [18].

### 2.2. How Much Lateralization to Improve ER and EXT?

LAT has been shown to increase ROM in several studies based on computer models from patient CT data [9,18,19]. The amount of LAT of the glenoid should probably be adapted to bring the remaining rotator cuff close to its original length and the cuff insertion points on the humerus closer to their original location [18,22,24]. Although this might seem to be the most logical target to aim for, this has not yet been proven to our knowledge. The change of LAT of the humeral tuberosities provided by planning software or manual measurements may be useful as an estimate during the radiologic planning process. We call the intraoperative process of finding the appropriate cuff tension “cuff balancing” [24], which is one of the challenges that we will discuss in part II: (V) “Lateralization, biomechanics and muscle length”. The amount of glenoid LAT can increase depending on the loss of preoperative bone stock. Arenas-Miquelez et al. concluded from their computer model study on Walch A1 glenoids [20] that only glenoid LAT has a significant effect on increasing the total global ROM and that a semi-inlay 145° model combined with 4 mm LAT and 2 mm inferior eccentricity provides the middle ground and most universal approach to RSA [19].

### 2.3. Glenosphere Size in the Context of Lateralization

Page and colleagues analyzed cumulative revision rates of reverse shoulder arthroplasty with data from the Australian National Joint Registry (ANJR) by looking at the influence of different glenosphere sizes. They found that GS < 38 mm may increase revision rates in primary RSAs, with significantly lower revision rates in females with 38–40 mm GS sizes and in males with > 40 mm GS sizes compared to GS sizes < 38 mm [25]. However, these registry findings were mostly based on Grammont-type RSA with a medialized COR implanted after 2004, a design in which a larger glenosphere size may counter weaknesses of Grammont’s design such as instability, insufficient posterior cuff tension and deltoid wrapping.

Haidamous and colleagues analyzed radiographic parameters of RSA influencing the range of motion outcomes in 160 patients with clinical follow-up out of a cohort of 200 [26]. They stratified their cohort into 36 patients with good ROM outcomes and 42 with poor ROM outcomes and concluded that a larger GS size (three sizes used: 36, 39 and 42), as well as inferior positioning, improved the range of motion following reverse shoulder arthroplasty. However, in all three of their statistical analyses, inferior glenosphere offset (IGO) was the most statistically significant parameter (*p* ≤ 0.002 in three different multivariate analyses), a parameter directly influenced by the GS size and baseplate size, which can be improved by GS eccentricity alone. GS eccentricity was only utilized in one of their patients. The strong correlation between combined lateralization of the baseplate with inferior glenosphere overhang (inferior offset) and improved ROM has recently been shown [18]. In the same study, a larger GS (39 mm) without glenoid lateralization was outperformed by a smaller GS with eccentricity (36 mm + 2 mm) with combined lateralization by maintaining a sufficient inferior overhang (Figure 2). As Haidamous and colleagues acknowledge in the discussion of the limitations of their study, the recommendation of a larger GS size cannot be generalized, should be patient-specific and depends on patient size and body height (not documented in their study) [26].

Ott and colleagues carried out a biomechanical cadaveric study testing the strain in three segments of the deltoid muscle (clavicular, acromial and spinal head) by implanting different RSA configurations in six cadavers [27]. They tested 0 mm, 5 mm and 10 mm glenoid-sided lateralization with GS sizes of 38 mm and 42 mm and found that the use of a 42 mm GS increased the strain in each segment significantly. The authors concluded that the lateralization and GS size significantly increased deltoid loading. According to the aforementioned priorities of combined glenoid lateralization and inferior GS overhang [18], the GS size should be adapted to patient height and size with caution to prevent overstuffing with too much strain on the deltoid origin.

### 2.4. External Rotation at 90° of Abduction

External rotation in abduction is a frequent physiological movement to reach the back of the head and to position one’s hand in space. An in silico study suggests that the best ER in ABD can be obtained with an optimized 145° semi-inlay design combined with baseplate LAT and an eccentric 36 mm GS [19]. Decreasing the NSA is powerful to increase EXT and rotation at 0° of ABD but to the detriment of glenohumeral motion at 90° of ABD. Conservation of the subacromial space has been shown to be important in several glenohumeral computer modeling studies with a static scapula. This may, however, be well compensated physiologically by scapulothoracic motion. This challenge will be discussed in detail in part II: (III) “Conservation of sufficient subacromial and coracohumeral space”.

### 2.5. Active ER

Restoration of active ER in patients with a preoperative ER lag with the arm at the side or at 90° elevation can be a major challenge for RSA in pseudoparalytic patients with a hornblower sign. One of the first multicenter studies of Grammont-type RSA showed that, in patients with no functional teres minor, the Constant score was significantly reduced (58 vs. 67 points, *p* < 0.01) [4]. Due to unsatisfactory results in this subset of patients treated by RSA alone, Boileau described a combined loss of active elevation and ER (CLEER), recommended simultaneous RSA and tendon transfer to address both the vertical and the horizontal imbalance of these shoulders and introduced a score for activities of daily living requiring ER (ADLER score, 30 points) [28]. A case series of 17 patients showed significant improvements after combined RSA with tendon transfer [29]. The mean preoperative ADLER score of the 17 patients was 7 (SD: 6), which was significantly improved to 25 (SD: 5) after a minimum follow-up of 12 months.

The differentiated evaluation of this condition is important [30]. Whether it is best treated by a simultaneous tendon transfer and RSA or by RSA with either glenoid-sided or humeral-sided lateralization remains controversial [31,32].

Berglund et al. retrospectively reviewed patient records after RSA with a lateralized COR to identify patients with CLEER-type pseudoparalysis. Out of 389 RSA, 24 patients were retrospectively identified to have presented a CLEER shoulder. ER was only assessed with the arm at the side. The ADLER score could not be evaluated, probably due to the retrospective study design. The 24 CLEER patients improved their mean active ER significantly (*p* < 0.01) from minus 21° to plus 28° [33]. A sub-analysis of 4 out of 24 patients with Goutallier stage 4 fatty infiltration of the teres minor showed a poorer mean active ER of 17° with the arm at the side. Functional ER with the arm in space in front of the body, which is indirectly measured by the ADLER score, was not evaluated.

Young et al. conducted a randomized controlled trial selecting 31 CLEER patients with a hornblower sign that were randomized for RSA with combined lateralization (minimal at the base plate and more on the humeral side) or RSA with a tendon transfer [11]. At 2 years, 10 patients remained for follow-up in the RSA alone group and 12 in the RSA with transfer group. A hornblower sign could be reversed in 73% of the cases in the RSA with transfer group and in 58% in the RSA alone group. However, the authors drew the conclusion that both RSA with or without transfer improved the ADLER score significantly. Looking at the mean preoperative ADLER score for both groups, it is remarkable that the score measured 17 points for both groups compared to the patients in Boileau’s study with a mean score of 7. It seems that Boileau’s patients had a more severe form of CLEER-type pseudoparalysis preoperatively. The patient selection, statistics and interpretation of the results of Young’s randomized trial were questioned in a letter to the editor [34].

Hamilton and co-workers published the biomechanical theory of an improved moment arm of a medialized glenoid–lateralized humerus (MGLH) RSA for the posterior deltoid in ER at 30°of ABD, as discussed in part II: (V) “Moment arms and muscle tensioning” (part II) [35]. Di Giacomo has most recently presented the results of a computer model showing improved deltoid fiber recruitment for ER at 60° of ABD with a RSA lateralized at the glenoid (LGMH) compared to RSA lateralized at the humerus (MGLH). He also presented a rotator cuff fiber length closer to the normal anatomy for LGMH RSA [36], which may be advantageous for the contractility of muscle fibers of the cuff and deltoid, according to the Blix curve [37] (Figure 4), as previously outlined for tendon transfers [38] and advantageous for active ER due to fiber recruitment. It remains unclear to date how to combine biomechanical RSA optimization of the moment arm and forces with the anatomical RSA optimization of fiber recruitment and length of contractile elements, according to Blix (Figure 4). The author of the present review uses a combination of glenoid-sided and humeral LAT with the aim to position the greater tuberosity at ±0 mm of LAT. The posterior deltoid may be a contributor to active ER after RSA. However, a tenodesis effect of the remaining posterior cuff may be essential. If the posterior cuff is completely absent, the posterior deltoid may be insufficient to restore active ER.

Patients with a persistent loss of ER and a hornblower sign after RSA may also be treated by a second stage tendon transfer if a function is unsatisfactory [39]. Improved FLEX and active ER were shown in 10 patients receiving a staged transfer after RSA with non-satisfactory ROM and ER. In our experience, patients with severe CLEER-type pseudoparalysis with high-grade fatty infiltration of the teres minor, a positive hornblower sign and an ER lag of more than 30° in position ER1 (arm at side) and ER2 (in 90° of ABD) are at high risk of unsatisfactory RSA function with ER weakness, persistent hornblower sign and a low ADLER score if treated with lateralized RSA alone. These patients may require treatment with an RSA and simultaneous tendon transfer if deemed fit and suitable for a more protected rehabilitation. A second-stage tendon transfer as published by Puskas et al. [39] may be a bail out but brings the inherent risk of a second procedure, which patients may fear or even refuse despite an unsatisfactory result.

## 3. Internal Rotation

Internal rotation (IR) is required for several activities of daily living. More specifically, buttoning a shirt, fastening a belt, dancing, getting dressed, bathing and toileting are difficult when IR is absent or reduced [40,41]. Traditional Grammont-style RSA is associated with reduced IR [42]. Medialization of the COR and loss of the deltoid wrapping angle may reduce the amount of anterior cuff that can be recruited for IR. A Grammont prosthesis also distalizes the humerus, increasing the tension and changing the line of the pull of the subscapularis tendon. For this and other reasons, some surgeons choose not to repair the subscapularis tendon. Furthermore, the subscapularis tendon may be of poor quality or irreparable, and failure of subscapularis repair is common [43,44,45].

Changes in prosthesis designs have tried to address this lack of IR. Huish et al. simulated RSA implantation using a three-dimensional model on 25 consecutive patients with Walch A1 glenoids [11]. Variations were made in the RSA parameters to assess the effect on IR. The largest gain in IR from a single parameter change was seen after distalization. Baseplate lateralization, increased GS diameter, increased humeral anteversion and varus NSA (135°) were all associated with improvements in IR, and the largest impact was seen by combining all these factors. This study, however, is a computer simulation only based on bony impingement, and it is clear that many other factors contribute to IR in vivo, such as soft tissue contractures and muscle function.

Langohr et al. conducted a cadaver study assessing the effect of the GS size on muscle force, joint load and motion [46]. They found reduced IR with a larger GS (42 mm), a finding they postulated was due to posterior capsular tension, as it forces the humerus to wrap around a large 42 mm GS in IR.

LAT has been shown to be associated with increased IR in clinical studies [47,48]. Werner et al. assessed 455 RSA patients at 1 year post-surgery [48]. They demonstrated improved IR at 90° of ABD in patients with greater glenoid LAT (8 mm performed better than 6, which was better than 0–4 mm). The assessment of functional IR by the vertebral level showed less difference between the groups, but there were still significantly more patients achieving a level of L4 or greater in the lateralized glenoid groups. Erickson et al. assessed 203 RSA patients at 2 years post-surgery [47]. They measured the overall LAT (including glenoid and humeral component) and found a small but statistically significant improvement in IR in patients with greater LAT. Increased BMI has also been shown to be an independent predictor of reduced IR in two clinical studies [49,50].

Changing the torsion of the humeral stem has been demonstrated to affect the ROM in numerous biomechanical studies [51,52,53,54]. Increased retrotorsion of the stem will generally increase ER but at the cost of IR. Similarly, increased antetorsion will increase impingement-free IR but at the cost of ER. Changing the humeral torsion, however, will also affect ROM in other planes and the sites of impingement, both intraarticular and extraarticular [51]. Berton et al. also demonstrated that humeral retrotorsion can increase the length and passive tension of the teres minor, which may paradoxically improve ER in RSA but decrease IR if the teres minor is overdistended [52] Similarly, changes in the length and tension of the subscapularis tendon may have an effect on IR.

A number of clinical studies have looked at subscapularis repair following RSA and its effect on IR. Dedy et al. assessed 48 shoulders that had undergone RSA through a deltopectoral approach with subscapularis repair [43]. Following ultrasonographic examination, 46% of the tendon repairs were graded as “intact” vs. 54% as “not intact”. The IR ability was scored on a 6-point ordinal scale based on the Constant score, with 1 being the lowest score (hand positioned at the lateral thigh) and 6 the highest score (hand placed behind back at the interscapular level). The “intact” tendon group achieved a median IR of 4 (interquartile range, 3–5), which was significantly higher than IR of the “not intact” group (median, 2; interquartile range, 1–3; *p* = 0.006).

Collin et al. performed a similar retrospective review of 86 patients who underwent primary RSA with subscapularis repair [45]. A postoperative ultrasound assessment at 2 years revealed that only 41 out of 78 patients (52.6%) had a healed subscapularis tendon. They demonstrated significantly better IR with no difference in ER (*p* < 0.01) in patients that had intact subscapularis tendon repairs vs. those with failed repairs. Both of these studies only included Grammont-type RSA; therefore, these results may not necessarily be generalized to newer, more lateralized designs. Other studies found no difference in IR when the subscapularis was repaired vs. not, but these trials did not radiologically assess the integrity of the repairs postoperatively [48,50].

More recently, a study by Gerber et al. showed the importance of functional extension for internal rotation to the back [10]. The authors pointed out that at least 40° of active extension is essential to achieve a functional IR allowing placement of the hand behind the back to reach the lumbar spine and any higher level. These clinical findings underline the importance of focusing on restoring sufficient extension during preoperative planning, as discussed for challenge (I): “External rotation and extension”.

In summary, improvements in IR may be achieved with more lateralized RSA prostheses, as well as a more inferiorized glenoid base plate with maximized inferior GS overhang. All these factors also improve the impingement-free extension as an essential requirement for functional IR. Decreasing the humeral retrotorsion may also improve the IR but at the cost of the ER. Studies have also demonstrated that a successful subscapularis repair can improve the IR, but the rate of healing was modest.

## 4. Conclusions

Optimizing passive external rotation, extension and internal rotation is the key to good function and the prevention of notching following RSA. The evidence on how to best restore active external and internal rotation is still inconclusive. Active internal rotation benefits from successful subscapularis repair, but lateralization alone may be sufficient. The quality of active external rotation post-RSA is affected by preoperative cuff tear patterns involving infraspinatus and the teres minor. Complementary factors are the conservation of sufficient subacromial and coracohumeral space, cuff tensioning, muscle length restoration with an optimized cuff and deltoid fiber recruitment, as well as biomechanics of the resulting moment arm. These additional challenges will be discussed in detail in part II.

## Figures and Tables

**Figure 1 jcm-12-01814-f001:**
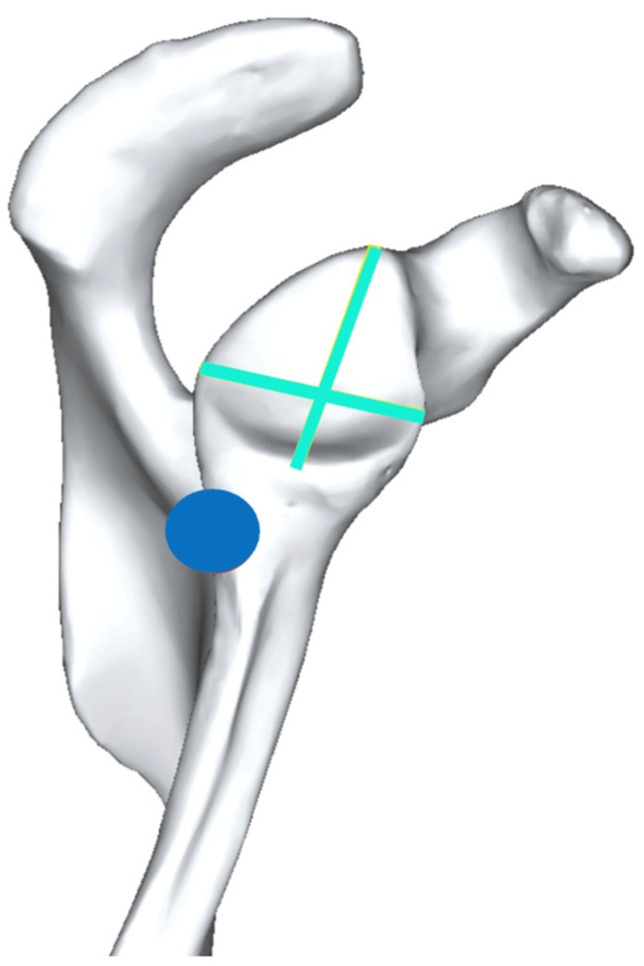
Posterior offset of the lateral border of the scapula in the glenoid plane.

**Figure 2 jcm-12-01814-f002:**
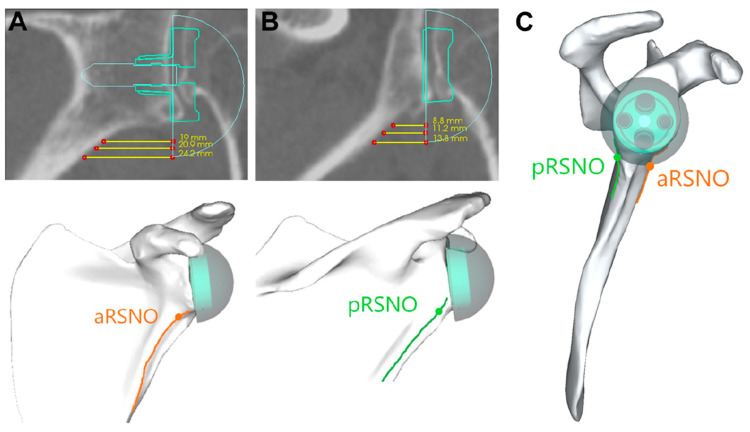
(**A**) Location of aRSNO in a sagittal view. (**B**) Location of pRSNO in a sagittal view. (**C**) Location of pRSNO and aRSNO in a coronal view. Figure reused with permission from the article of Bauer et al. [18]. aRSNO: anteroinferior relevant scapular neck offset; pRSNO: posteroinferior relevant scapular neck offset.

**Figure 3 jcm-12-01814-f003:**
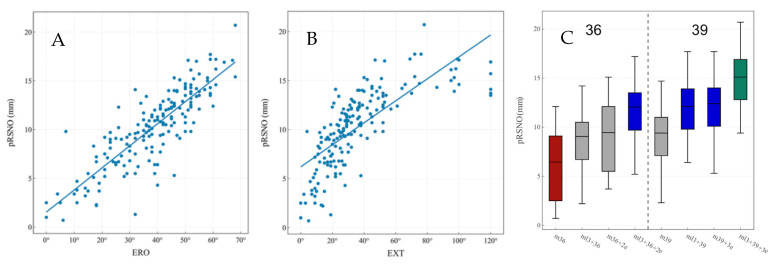
(**A**) Correlation between pRSNO and ERO. (**B**) Correlation between pRSNO and EXT. (**C**) Effect of glenosphere type on pRSNO. Figure reused with permission from the article of Bauer et al. [18]. aRSNO: anteroinferior relevant scapular neck offset; pRSNO: posteroinferior relevant scapular neck offset; ERO: external rotation; EXT: extension.

**Figure 4 jcm-12-01814-f004:**
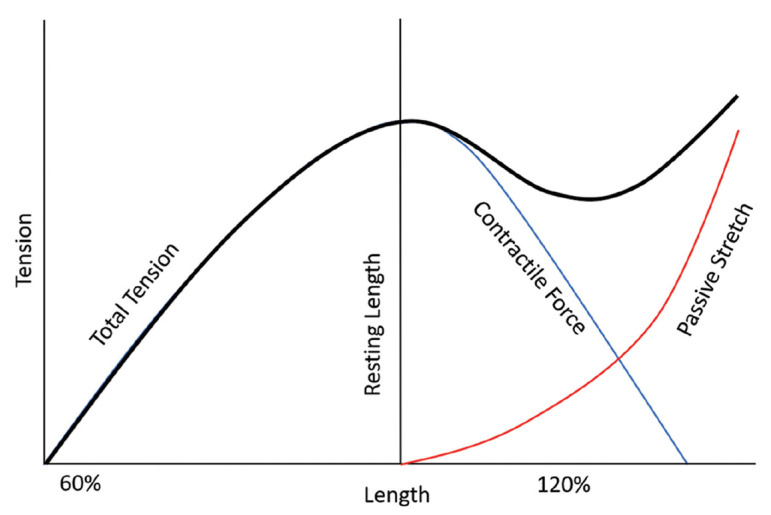
The Blix curve is the summation of two tension-length curves: one for active contraction and one for elastic recoil against the length. It demonstrates the greatest contractile force when stimulated at the resting length with a plateau of some length–distance change before the contractile force drops off. Figure reused from the article of Gardenier et al. [37].

## Data Availability

Not applicable.

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
