# Peer review of "Challenges for Optimization of Reverse Shoulder Arthroplasty Part I: External Rotation, Extension and Internal Rotation"

_jcm, 2023, doi:10.3390/jcm12051814_

Round 1
Reviewer 1 Report
The manuscript is a review on "Challenges for Optimization of Reverse Shoulder Arthroplasty Part I: External Rotation, Extension and Internal Rotation" .The manuscript is well written and the summary of the review may be used as an aide memoire for RSA planning.
The following suggestion may be incorporated for better technical appeal and readbility of the manuscript.
1. the introduction should be more precise on why this work is taken up. include a paragraph towards the end of the introduction about the novelty and outcome of the current review.
2. The conclusions should be re-written.
3. Seems like figure 1 has no reference. please inlcude the reference.
4. Figures 2 & 3 are taken form same reference. Suggest to include more figures from different references.
Reviewer 2 Report
Dear author,
it is an extensive research performed, clearly presented and with pertinent conclusions. Some small mis-spellings must be corrected, I suggested a few.
So it can be published after minor language corrections.
Best wishes!

Reviewer 3 Report
The authors conduct a generally well-written, fairly comprehensive review of the considerations related to achieving internal and external rotation with reverse shoulder arthroplasty. However, several changes are needed and some of the more speculative statements need to be removed before this paper can be published.
On line 59, the authors should provide a references for this statement: “Notching has been shown to be associated with adverse clinical outcomes” Note that reference 2 is not appropriate for that portion of the sentence. Existing references 14 and 15 are probably best. Related to reference 2, I don’t think that is really the best reference for the second portion of the sentence. I think the correct reference for inferiorly shifting the baseplate to avoid notching is Nyffeler et al JSES 2005, as follows: Nyffeler RW, et al. Biomechanical relevance of glenoid component positioning in the reverse Delta III total shoulder prosthesis. J Shoulder Elbow Surg. 2005 Sep-Oct;14(5):524-8.
This statement starting on line 108-111 (“Werthel et al. point out that humeral LAT cannot reduce notching. However, although decreasing the NSA to 145° or 135° has very minimal influence on humeral LAT, it does reduce the risk of notching by tilting the polyethylene insert away from the scapular pillar”) needs to be modified as reference 22 is not who first described these considerations. You can still include it, but there are many other studies from a decade or more before which probably should be referenced instead, here is a few (there are others as well):
Roche C, et al. Geometric analysis of the Grammont reverse shoulder prosthesis: an evaluation of the relationship between prosthetic design parameters and clinical failure modes. Proceedings of the 2006 ISTA Meeting. 2006.
Gutiérrez S, et al. Center of rotation affects abduction range of motion of reverse shoulder arthroplasty. Clin Orthop Relat Res. 2007 May;458:78-82.
Gutiérrez S, et al. Range of impingement-free abduction and adduction deficit after reverse shoulder arthroplasty. Hierarchy of surgical and implant-design-related factors. J Bone Joint Surg Am. 2008 Dec;90(12):2606-15.
Roche C, et al. An evaluation of the relationships between reverse shoulder design parameters and range of motion, impingement, and stability. J Shoulder Elbow Surg 2009;18(5)
De Wilde LF, et al. Prosthetic overhang is the most effective way to prevent scapular conflict in a reverse total shoulder prosthesis. Acta Orthop. 2010 Dec;81(6):719-26.
Lines 119-138 don’t add much to the paper and are written too densely. I would recommend just deleting these lines entirely to help the flow of the work.
The entire section starting in lines 155 needs to be rewritten, its seems somewhat biased to the authors particular perspective as opposed to a broader summary experience from the literature. There are many rTSA systems which provide multiple diameters of glenospheres which are generally selected based on a patient’s size.(i.e. larger diameter used with larger anatomy) And there are many patients who have achieved positive clinical outcomes with larger size glenospheres (i.e. 40mm and above). A simple review of the literature would demonstrate that, there is nothing in the literature related to this section that suggests that a given range of glenospheres diameters is optimal. Accordingly, this section needs to be rewritten more broadly and better referenced.
The statement in lines 159-160 (“A larger GS may be associated too much overall lateralization which may lead to pain, loss of function and even stress fractures of the scapular spine.”) needs to be deleted. The referenced study is also misquoted as it relates to scapular spine fractures – larger glenosphere implies bigger diameter, which was not the reported finding of that reference - that particular study found lateralized CoR are associated with greater scapular spine stresses because of the reduced deltoid mechanical advantage…a thinner glenosphere relative to the diameter is the relevant variable that defines that increased lateral offset.
Line 167-168 is worded imprecisely: “A middle sized hemispherical glenosphere”. What does middle sized mean? The authors need to be specific. Whatever specific language the authors decide, they should also use it to correct the same “middle sized” reference in line 287.
The internal rotation section needs to be re-written and refined by the authors some, this is editorial, but it just plods along without succinctly making the points, as the authors did nicely in the conclusion section (i.e. lines 354-360). To tighten this section up, I would recommend just deleting lines 283-291 - they don’t provide much value and have little clinical relevance. Similarly, lines 311-343 need to be tightened up, there are too few points made for how many lines there are. Perhaps just delete lines 320-333?
Aside from these points, I think that this is a well done review of relevant rTSA parameters associated with internal and external rotation. Thank you to the authors for this work.
